# MicroRNA in Acromegaly: Involvement in the Pathogenesis and in the Response to First-Generation Somatostatin Receptor Ligands

**DOI:** 10.3390/ijms23158653

**Published:** 2022-08-04

**Authors:** Daniel G. Henriques, Elisa B. Lamback, Romulo S. Dezonne, Leandro Kasuki, Monica R. Gadelha

**Affiliations:** 1Neuroendocrinology Research Center, Endocrinology Division, Medical School and Hospital Universitário Clementino Fraga Filho, Universidade Federal do Rio de Janeiro, Rio de Janeiro 21941-901, Brazil; 2Neuropathology and Molecular Genetics Laboratory, Instituto Estadual do Cérebro Paulo Niemeyer, Rio de Janeiro 20231-092, Brazil; 3Neuroendocrinology Division, Instituto Estadual do Cérebro Paulo Niemeyer, Rio de Janeiro 20231-092, Brazil; 4Endocrinology Division, Hospital Federal de Bonsucesso, Rio de Janeiro 21041-020, Brazil

**Keywords:** acromegaly, miRNA, tumorigenesis, somatostatin receptor ligands

## Abstract

Acromegaly is a chronic and systemic disease due to excessive growth hormone and insulin-like growth factor type I caused, in the vast majority of cases, by a GH-secreting pituitary adenoma. About 40% of these tumors have somatic mutations in the *stimulatory* *G protein alpha-subunit 1* gene. The pathogenesis of the remaining tumors, however, is still not fully comprehended. Surgery is the first-line therapy for these tumors, and first-generation somatostatin receptor ligands (fg-SRL) are the most prescribed medications in patients who are not cured by surgery. MicroRNAs are small, non-coding RNAs that control the translation of many mRNAs, and are involved in the post-transcriptional regulation of gene expression. Differentially expressed miRNAs can explain differences in the pathogenesis of acromegaly and tumor resistance. In this review, we focus on the most validated miRNAs, which are mainly involved in acromegaly’s tumorigenesis and fg-SRL resistance, as well as in circulating miRNAs in acromegaly.

## 1. Introduction

Acromegaly is a chronic and systemic disease due to excessive growth hormone (GH) and insulin-like growth factor type I (IGF-I) levels. It is a rare disease, with an estimated prevalence of 40 cases/1,000,000 inhabitants and annual incidence of 3–4 new cases/1,000,000 inhabitants [1]. Patients with acromegaly present with an enlargement of extremities, arthralgia, diabetes mellitus, arterial hypertension, neoplasias, and obstructive sleep apnea, among other complications [2,3]. Compressive symptoms due to tumor growth beyond the sella turcica can be seen, and lead to visual field defects and panhypopituitarism [4]. 

Surgery is the treatment of choice [5]. In specialized centers, cure is attained in 80% of microadenomas cases and 40–70% of macroadenomas [6,7]. In uncured patients, octreotide or lanreotide, which are first-generation somatostatin receptor ligands (fg-SRL), are the most prescribed medications [6]. GH-secreting pituitary adenomas express four types of somatostatin receptors: SST1, SST2, SST3, and SST5 [8,9,10]. Fg-SRL preferentially bind to SST2 [10,11]. Its efficacy in controlling the disease at the biochemical level ranges from 30 to 40% [10,11]. Pasireotide is a second-generation SRL with a high affinity for SST5, followed by SST2 [12]. Disease control with pasireotide is seen in at least 20% of uncontrolled patients with fg-SRL [13,14,15,16].

Acromegaly is caused, in the vast majority of cases, by a sporadic GH-secreting pituitary adenoma [4]. About 40% of these tumors have somatic mutations in the *stimulatory G protein alpha-subunit 1* (*GNAS1*) gene [17,18]. Germline mutations in the *aryl hydrocarbon receptor-interacting protein* (*AIP*) gene can be found in approximately 4 to 12% of patients with apparently sporadic acromegaly, especially if they are young (age < 30 years old) and carry macroadenomas [19,20,21,22]. Apart from somatic and germline mutations, other mechanisms, such as epigenetic changes leading to abnormal gene expression, can be observed in the context of acromegaly [23]. Differentially expressed microRNAs (miRNAs) in tumor tissue, as well as circulating miRNAs, have been described in patients with acromegaly [23]. The pathogenesis of acromegaly and tumor resistance to treatment can be associated with differently expressed miRNAs [23]. In the present review, we will briefly address the current knowledge regarding miRNAs’ mechanism of action and the role of these molecules as biomarkers in pathological processes. In the second part, we will review the miRNAs associated with acromegaly tumorigenesis and resistance to fg-SRL, as well as circulating miRNAs in acromegaly.

## 2. miRNAs: Definition and Mechanism of Action

miRNAs, first described by Ambros and co-workers [24], are small, non-coding RNAs of approximately 22 nucleotides (nt) that act as post-transcriptional repressors of target mRNAs in eukaryotic cells. miRNAs are largely responsible for important cellular responses to signaling and stress [24]. A great variety of miRNAs have been discovered, and it is predicted that 1 to 5% of the human genome is composed of miRNA genes. Furthermore, it is estimated that they regulate about 30–60% of all protein-codifying genes, constituting a major class of molecular regulators [25,26]. The discovery of miRNAs in 1993 was a hallmark in molecular biology, changing the classical dogma [24]. Many studies have been dedicated to these new molecules [24].

miRNAs are generated from a classical pathway, as well as by alternative biogenesis pathways known as non-canonical pathways. In the classical pathway, miRNAs are mainly transcribed by RNA polymerase II in a hairpin structure known as pri-miRNA [27]. Furthermore, few miRNAs are transcribed by RNA polymerase III [26]. Following this transcription, nuclear processing occurs with a microprocessor complex, which includes an enzyme called DROSHA (an RNAse III endonuclease), DiGeorge syndrome region 8 subunit (DGCR8; an RNA-binding protein), and other proteins [27]. This results in a ~70-nucleotide stem-loop pre-miRNA, which is exported from the nucleus to the cytoplasm by Ran-GTPase exportin 5. In the cytoplasm, they are processed by another RNA endonuclease III enzyme, named Dicer, which removes the terminal loop of the sequence, generating a double-strand, 19–25-nucleotide miRNA. Afterwards, this miRNA is incorporated into the RNA-inducing silencing complex (RISC) with an Argonaute protein (AGO) that retains only one strand (5p or 3p), with this strand being the mature miRNA [28,29]. The miRNA associated with RISC binds to a complementary sequence in the 3’ untranslated region (3’-UTR) of the mRNA, leading to its degradation or translation repression depending on the complementarity level [24] (Figure 1). 

There are non-classical biosynthetic pathways, which are known as DROSHA-independent and Dicer-independent pathways [30]. A well-known DROSHA-independent pathway is, interestingly, known as “mirtrons” because it refers to certain introns that work as pre-miRNA and can be exported from the nucleus for further processing [31]. Few miRNAs are produced by the DICER-independent pathway, which needs the catalytic activity provided by isoform two of the Argonaute protein (AGO2) [32].

miRNAs control genic expression, usually by repressing translation. However, how this regulation occurs depends on factors in the cellular and biomolecular context. The specificity of the RISC complex acting on the target depends on the seed sequence, which is a 7nt sequence, usually in miRNA 5’ end. This must pair perfectly with its target in the miRNA response element (MRE), located in the mRNA. Moreover, the complementarity of MRE with the seed sequence indicates whether the mRNA will be degraded or whether translation will be repressed [33]. Although the most well-known miRNA action is translation suppression, miRNAs can also activate the transcription of a few genes, as previously described [33].

miRNAs are a major field of study in oncology, and their role in tumorigenesis and tumor suppression has been demonstrated. Some miRNAs have already been described as tumor suppressors, as they inhibit important protein-coding oncogenes [32]. For instance, the Kirsten rat sarcoma viral oncogene homolog (*KRAS*) proto-oncogene can be regulated by miR-143, miR-145, and the miR-200 family [34]. Furthermore, studies have described a group of miRNAs as oncomiRs because they inhibit tumor suppressors [35]. A known example of oncomiR is miR-21, which is upregulated and has an anti-apoptotic role in different tumors [34]. In addition, some miRNAs act in resistance to antineoplastic medications. The main mechanism is the regulation of proteins that are drug targets [32]. miRNAs can also directly or indirectly regulate genes that control cell efflux of medications and metabolic liver enzymes [32]. 

## 3. miRNA as Circulating Biomarkers of Disease

In recent years, miRNAs have been implied to be biomarkers of certain diseases, as they are altered in several pathological events, and can be released by apoptotic and necrotic cells [36,37]. Although miRNAs have an intracellular function, they have already been identified in several biological fluids, such as cerebrospinal fluid [38], urine [39], plasma [40] and serum [41], gingival fluid, and saliva [42]. Under these conditions, circulating miRNAs may be associated with proteins [43], microvesicles, or lipoprotein complexes [44]. The role of circulating miRNAs is not well-established at present, but studies have shown that miRNAs transmitted by exosomes [45] or HDL proteins [44] act on receptors in targeted distant cells, suggesting the involvement of these molecules in intercellular communication (Figure 2). Exosomes containing miRNAs are actively released by cells enabling long-distance communication among different cell types, and thereby regulating the gene expression of a distant target cell [44,46] and controlling metabolism [47].

The first findings show that circulating miRNAs may be used as biomarkers, as their tissues correlate; patients with B-cell lymphoma had higher miR-155, miR-210, and miR-21 serum levels than healthy individuals [41]. In the same year, another pioneer study described miRNAs in human blood, using plasma obtained from placenta to characterize and attest to their stability [48].

Mature miRNAs found inside cells have a remarkably long half-life and, in the extracellular environment, they are highly stable and resistant to degradation [49,50]. In serum, they are considerably stable for long periods of time, which can be explained by two mechanisms: the formation of the ribonucleoprotein complex together with AGO2 [43] and/or by incorporation into exosomes [45]. These properties give these molecules unique characteristics, such as: resistance to extreme variations in pH and temperature and repeated cycles of freezing and thawing and storage for long periods of time, without degradation [51]. Since they can be obtained through minimally invasive methods, in most different biological fluids, they are helpful biomarkers in different types of disease, especially neoplasms [40,51,52]. Furthermore, this evidence indicates that they have the potential to be used as less invasive biomarkers in early diagnosis, prognosis, and as candidates for therapeutic interventions [53].

## 4. miRNAs in Acromegaly

miRNAs have been a major area of study in acromegaly for the last two decades, with a wide variety showing some role in tumorigenesis and/or resistance to treatment, mainly resistance to fg-SRL. Additionally, they may also be used as biomarkers. 

### 4.1. miRNAs Involved in Tumorigenesis

Studies have revealed some miRNAs that are involved in somatotropinomas tumorigenesis, participating in important signaling pathways and being correlated with tumor proliferation, invasion, and size (Table 1).

Different miRNAs that potentially regulate the genes involved in GH-secreting adenoma tumorigenesis have been described as being up- or downregulated in these tumors [28]. In 2010, a study performed by microarray technique showed that 25 somatotropinomas had 52 differentially expressed miRNAs compared to the normal pituitary [55]. Some of these miRNAs were thought to be involved in tumorigenesis; however, not all were confirmed by quantitative polymerase chain reaction (qPCR). Of the confirmed ones, miR-126 is known to inhibit the regulatory beta subunit of phosphatidylinositol 3-kinase (PI3K) translation in colon epithelium. PI3K/AKT/mTOR pathway has been previously correlated with pituitary adenomas [59]. MiR-381 was also shown to be downregulated [55]. Both miRNAs were found to target the pituitary tumor-transformer gene 1 (PTTG1) protein, which can function as an oncogene in endocrine tumors when overexpressed [60]. PTTG1 shows high expression in GH-secreting pituitary adenomas compared to healthy pituitary glands and regulates tumor-related metastasis and therapeutic responses, promoting cell migration and proliferation, as well as suppressing cell apoptosis, in several tumors [61].

Mir-381 is also known to act as a tumor suppressor in different cancers; for example, it is downregulated in triple-negative breast cancer [62], lung adenocarcinomas [63], and others [64]. One of its targets is the *IGF-1R* gene, which is involved in AKT and ERK signaling pathways, regulating cell growth, apoptosis, migration, invasion, and drug resistance [64,65]. Another important target is nicotinamide phosphoribosyltransferase (NAMPT), which catalyzes reactions to produce NAD, contributing to DNA repair and the cell cycle. This enzyme is overexpressed in some cancers such as breast, colon, and gastric cancers [64]. 

Another study comparing GH-secreting pituitary adenomas to normal pituitary samples found eighteen downregulated miRNAs and one upregulated miRNA [57]. Some of these downregulated miRNAs were found to control high-motility group A 1 (*HMGA1*) (miR-34b and miR-548c-3p), *HMGA2* (miR-34b, miR-326, miR-432, miR-548c-3p, and miR-570), and E2 promoter binding factor 1 (*E2F1*) (miR-326 and miR-603) expression. These three genes, when overexpressed, have been associated with proliferation [57]. Moreover, high levels of HMGA2 have been correlated with tumor invasion, tumor size, and high Ki-67 in pituitary adenomas [66]. Therefore, it is suggested that these miRNAs play a critical role in GH-secreting pituitary adenoma tumorigenesis. 

A study, published in 2013, performed a microarray in twelve GH-secreting tumors and, compared to normal pituitaries, which found five upregulated miRNAs and twelve downregulated miRNAs, showed that the top two most upregulated miRNAs were miR-26b and miR-212 and the top two downregulated miRNAs were let-7a3 and miR-128 [56]. In the same study, the authors found that the inhibition of miR-26a and miR-26b and overexpression of miR-128 and miR-186 suppressed the invasiveness of an ACTH-secreting cell line from mice pituitary gland, Att-20 cells, in vitro. Apart from these results, it was also observed that the inhibition of miR-26b and overexpression of miR-128 blocked GH3 cell (a GH and prolactin cell line derived from pituitary tumor in rat) ability to form colonies and invade through PTEN and the polycomb group family of proteins BMI1 [56]. Additionally, miR-26b was shown to directly repress PTEN/AKT signaling, being overexpressed in GH-secreting pituitary adenomas [56,67]. MiR-26b and miR-128 act in the PTEN/PI3K/AKT pathway. PTEN downregulation can lead to phosphorylation mediated by AKT and the activation of nuclear factor kappa-B (NF-kB) activity, leading to p53 degradation, which plays a central role as a cell cycle suppressor [68]. Furthermore, BMI1 can inhibit the PTEN/PI3K/AKT pathway and has been identified as a proto-oncogene in other cancers [69]. Moreover, the PTEN/PI3K/AKT stimulates epithelial–mesenchymal transition (EMT) [70,71].

MiR-503 was also found to be downregulated in GH-secreting adenomas [55]. This miRNA is also known as a tumor suppressor, as it is negatively correlated with different kinds of cancer, targeting *IGF-1R*, genes that are involved in the EMT, as well as fibroblast growth factor 2 (*FGF2*), which plays a role in differentiation and proliferation [72].

Another miRNA that was found to inhibit EMT is miR-525-5p, which is upregulated in GH-secreting pituitary adenomas compared to normal pituitary [55]. This miRNA has been studied as a tumor suppressor in gliomas by targeting the STAT-1 signaling pathway that leads to proliferation, migration, invasion, and EMT [73]. It also inhibits anchorage-independent growth, resistance to apoptosis by loss of attachment to the extracellular matrix (anoikis), and partial tumor metastasis by inactivating ubiquitin conjugating enzyme E2 C/zinc-finger E-box binding homeobox 1/2 (UBE2C/ZEB1/2) signaling in cervical cancer [74]. This mechanism might suggest that this miRNA could be a protector against invasion, proliferation, and EMT when upregulated in GH-secreting pituitary adenomas.

Concerning tumor size, miR-16-1 was found to be downregulated in GH-secreting tumors by Northern blotting. A negative correlation was observed between the expression of this miRNA and the tumor diameter [58,75]. MiR-16-1 showed a negative correlation with arginyl t-RNA synthetase (*RARS*) mRNA expression, which is inversely correlated with p43 secretion [58]. P43 is thought to have antineoplastic activity [58], which suggests that miR-16-1 could be involved in protecting pituitary adenomas from tumorigenesis or tumor growth.

Furthermore, nine miRNAs were found to be differentially expressed in a microarray assay when comparing micro- and macroadenomas; however, only one upregulated miRNA, miR-524-5p, and one downregulated miRNA, miR-124, were confirmed by qPCR [55]. MiR-524-5p seems to be important in the pathogenesis of other tumors. For instance, in gliomas, miR-524-5p targets the expression of two proteins, Jagged-1 and transcription factor hairy, and is the enhancer of split 1 (Hes-1), components of the Notch pathway, which is an important signaling pathway and is highly conserved in cells determining response to cell hypoxia and angiogenesis [76]. In pituitary adenomas, the Notch pathway is still poorly understood; however, it seems to lead to more aggressive and invasive tumors [77]. Therefore, miR-524-5p might act as a tumor suppressor by interfering in growth and invasiveness in somatotropinomas. Furthermore, miR-524-5p has been studied in breast cancer to have an effect on inhibiting the follistatin-like 1 (*FSTL1*) gene, which is known to regulate cell proliferation, apoptosis, and metabolism [78]. Moreover, this miRNA was found to have an antiangiogenic effect by targeting a serine/threonine-protein kinase known as lysine deficient protein kinase 1 (*WNK1*), and also inhibiting cell proliferation and inducing cell cycle arrest in colon cancer [79]. Both pathways are poorly comprehended in pituitary adenomas; however, they might be a potential field of study. MiR-124 is also associated with many different diseases; however, in malignant diseases, it plays a role by inhibiting Caspase-3 and -9 activation, WNT/B-catenin pathway, PI3K/AKT and MAPK/ERK pathways, and was shown to be downregulated in a great variety of cancers [80]. MiR-524-5p and miR-124 seem to be involved in important pathways that could protect GH-secreting adenomas from being more aggressive.

MiR-338-3p has also been correlated with GH-secreting pituitary adenomas, and was shown to be upregulated in invasive tumors [54]. Additionally, when miR-338-3p was inhibited in GH3 cells, their proliferation rate was lower [54]. Furthermore, a high expression of miR-338-3p increases PTTG1 levels in vitro, whereas immunofluorescence for PTTG1 showed more stained cells in tumors [54]. Even though the study did not identify the exact target of miR-338-3p, which leads to a PTTG1 upregulation, this gene is associated with different cancers, leading to growth, migration, and EMT [81,82]. Furthermore, it was seen that PTTG1 expression was higher in invasive pituitary tumors [83]. Finally, miR-107 showed a negative correlation with AIP in vitro; however, this was not demonstrated in vivo [84]. *AIP* is an important tumor suppressor that controls the cell cycle through p53-p21/p27 pathway and is a predictor of response to fg-SRL in acromegaly [85].

The molecular and cellular mechanisms by which miRNAs are involved in tumorigenesis are not fully known. Therefore, understanding how miRNAs are involved in tumorigenesis, cell signaling, and interactions can help to develop therapeutic approaches and early diagnosis, and to establish possible biomarkers.

### 4.2. miRNAs Involved in Treatment Resistance

In acromegaly, as seen in other tumors, miRNAs also play a role in resistance to treatment by different mechanisms, which some studies have tried to elucidate (Table 2). For instance, miR-34a was shown to be correlated to AIP expression, which is an important marker of response to fg-SRL [86,87]. Even though both studies showed miR-34a’s importance, they reached this conclusion using different approaches. The first study, after excluding *AIP*-mutated patients, showed that AIP levels, in immunohistochemistry, were negatively correlated with miR-34a in GH-secreting pituitary adenomas. Furthermore, patients with lower miR-34a levels were more responsive to fg-SRL therapy than patients with higher mir-34a. In this context, it was shown that AIP is a target of miR-34a. Therefore, depending on the level of this miRNA, the tumor has a different response to fg-SRL [86]. However, Bogner et al. showed complementary data, suggesting that AIP regulates miR-34a expression, which targets alfa subunit of G inhibitory protein (Gαi2) from SST2, which results in a poor response to fg-SRL [87]. Regarding proliferation and apoptosis, miR-34a was found to promote the proliferation and clonogenicity of GH-secreting pituitary adenoma cells, as well as the suppression of apoptosis [87]. 

An important predictor of response to fg-SRL is SST2. The presence of SST2 expression is essential to the response to first-line pharmacological treatment [88]. Therefore, miRNAs that interfere with SST2 expression can impact the response to fg-SRL (Figure 3). The receptor was found to be regulated by miR-185 in vitro by binding in position 3612-3619 of SST2 3’ UTR region, and this miRNA was found to be upregulated in non-responders in vivo [89]. However, no direct correlation was observed between mir-185 and SST2 in vivo. Patients were pretreated with lanreotide for four months prior to surgery, which can reduce SST2 expression and may have altered the expression of this protein in immunohistochemistry [89]. 

The factors that interfere in fg-SRL response, however, are not completely understood. Many signaling pathways may play a role in the medication outcomes, suggesting that miRNAs are involved [23]. For instance, a study showed that 13 miRNAs were differentially expressed when patients were pretreated with lanreotide, and seven miRNAs were differentially expressed when comparing fg-SRL responders with non-responders [55]. Of those involved in the fg-SRL treatment response, miR-125a-5p and miR-524-5p are downregulated in responders to treatment and they have been correlated with insulin-like growth factor binding protein 3 (IGFBP-3) and insulin-like growth factor binding protein acid labile subunit (IGFALS) chain precursor, two proteins involved in cell communication, growth regulation, and receptor binding [55]. MiR-524-5p is downregulated in fg-SRL responders and targets matrix metalloproteinase-9 (MMP-9), which is involved in metabolism, ion binding, and extracellular matrix remodeling [55]. Another miRNA that is upregulated in responsive GH-secreting pituitary adenomas is miR-886-5p [55]. Moreover, miR-886-5p has been considered an oncomiR, since it is overexpressed in high-grade and invasive bladder tumors [90] and cervical cancer cells [91]. Additionally, in multiple myeloma cells, miR-886-5p triggers proliferation and inhibits p53-Bax signaling pathway, which leads to the suppression of apoptosis [92]. Apart from being related to aggressiveness in different cancers, miR-886-5p might be a useful biomarker of response to treatment in somatotropinomas.

Efforts to apply personalized medicine in acromegaly have been made in recent years and the search for predictors of response is a field in progress [93]. More studies should be performed to establish miRNAs as a useful tool in pharmacological treatment in acromegaly.

## 5. Circulating miRNAs in Acromegaly 

Circulating miRNAs are being researched as biomarkers in acromegaly to find an easy and less invasive approach to detect disease, tumor growth, and other characteristics. Recently, comparing healthy controls and patients with somatotropinomas, miR-29c-3p was downregulated in patients with acromegaly, and also in patients with poor biochemical control using fg-SRL [94]. According to targetscan, this miRNA is known to target over 1200 mRNAs (targetscan.org) and was shown to be a tumor suppressor that is downregulated in different neoplasias [94]. 

Another study found downregulated plasma miR-4446-3p and miR-215-5p in patients with acromegaly compared to healthy subjects [95]. Both miRNAs were correlated with IGF-I levels, suggesting its association with biochemical features. These circulating miRNA’s origin is unknown, and its overexpression may only be a consequence of high GH and IGF-I levels rather than being secreted from the pituitary adenoma [95]. MiR-215-5p is downregulated in colorectal cancer tissues, possibly acting as a tumor suppressor [96]. As patients with acromegaly have an increased colorectal cancer risk [4,97], miR-215-5p could be a target for further research in this context.

In a recent exosome miRNA sequencing, 169 differently expressed miRNAs were found when comparing six serum samples from patients with GH-secreting pituitary adenomas and six samples from healthy individuals; however, during the validation step, only miR-320a and miR-423-5p showed statistical significance, and the same tendency was shown in both exosome miRNA sequencing and qPCR [61]. MiR-320a and miR-423-5p were lower in the serum of samples from patients with GH-secreting pituitary adenoma than in the control sample [61]. MiR-423-5p inhibited the expression of PTTG1 [61]. The authors found that the increased expression of PTTG in GH-secreting pituitary adenoma could explain tumoral migration and proliferation, suggesting that miR-423-5p could be used as a therapy in GH-secreting pituitary adenomas in the future.

## 6. Conclusions

Acromegaly caused by GH-secreting pituitary adenomas can be associated with differently expressed miRNAs that might influence tumor proliferation, invasion, size, and response to fg-SRL. In this context, some miRNAs function as tumor suppressors, preventing tumor invasion and aggressiveness. On the other hand, oncomiRs target important cell cycle regulators. Furthermore, miRNAs might be differentially expressed when comparing micro- and macroadenomas or fg-SRL responders and non-responders. Additionally, in recent years, circulating miRNAs have been used as non-invasive biomarkers, although there are still no validated biomarkers that accurately predict the presence of a GH-secreting pituitary adenoma or its behavior. 

In future, miRNAs might be a useful tool for increasing the understanding of the pathogenesis of GH-secreting adenomas, and may be used in the development of new treatments, improving the quality of life of patients with acromegaly. As miRNAs usually bind to complementary sequence motifs in target RNA and, consequently, silence target genes, it would be interesting to use miRNA to silence genes related to pituitary tumorigenesis.

## Figures and Tables

**Figure 1 ijms-23-08653-f001:**
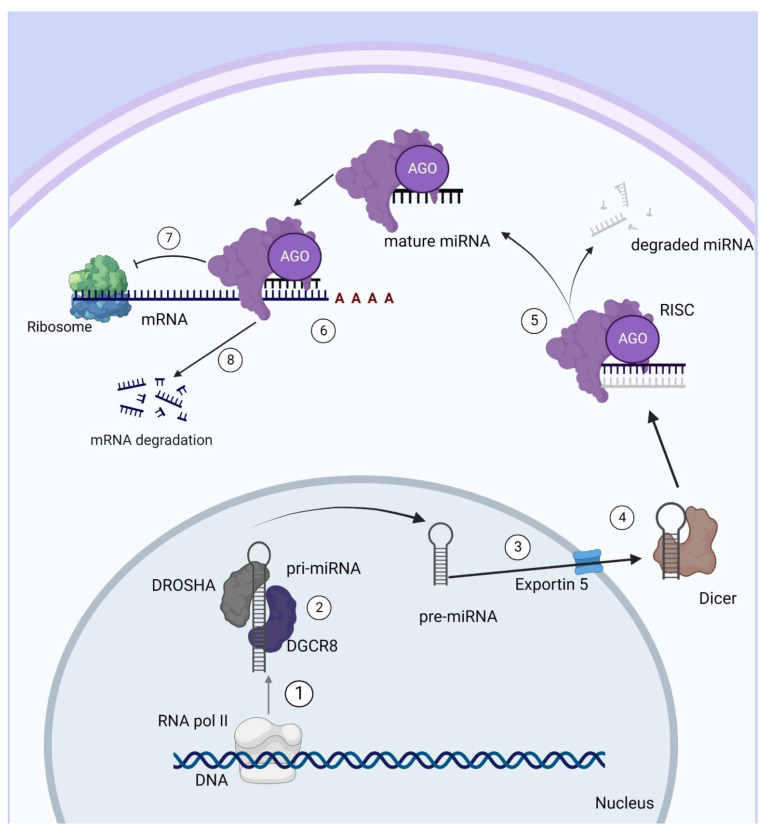
Classical pathway: miRNA synthesis. (1) miRNA is mostly transcribed by RNA polymerase II through genomic regions. (2) pri-miRNA, a hairpin sequence, is then processed by DROSHA and DGCR8 proteins, which results in pre-miRNA. (3) pre-miRNA is transported to the cytoplasm through Exportin 5. (4) In the cytoplasm, pre-miRNA is processed by Dicer, which removes the terminal loop generating a 19-25nt miRNA. (5) miRNA is incorporated into the RNA-inducing silencing complex (RISC) with Argonaute (AGO) protein that retains one of the strands, the mature miRNA, while the other strand is degraded. (6) Mature miRNA associated with AGO and RISC binds its seed sequence in the target miRNA response element (MRE) in the mRNA. After binding, mRNA translation might be inhibited by two main mechanisms. (7) RISC represses translation or (8) RISC leads to mRNA degradation.

**Figure 2 ijms-23-08653-f002:**
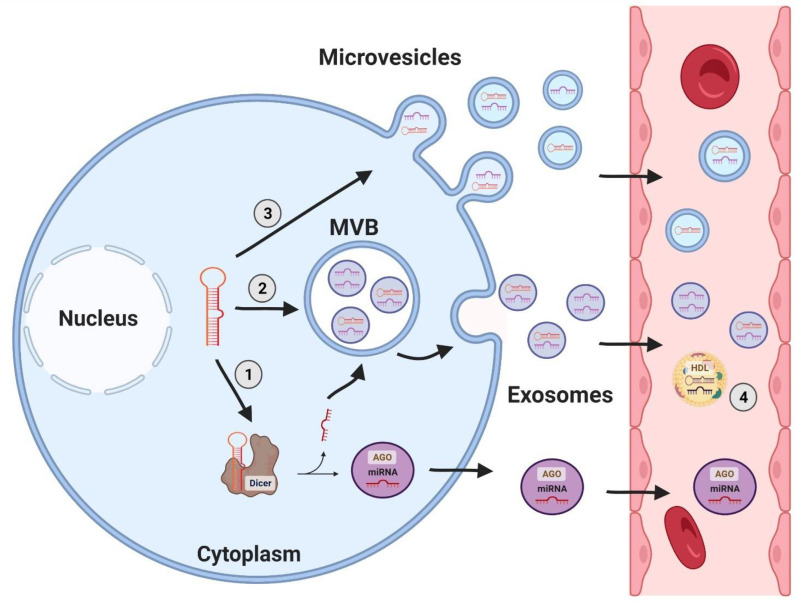
Circulating miRNAs. Circulating miRNAs arise in different ways: (1) Pre-miRNAs leave the nucleus and enter the classical processing pathway, where they might be secreted in their free form, by an unknown route, and circulate associated with high-density lipoprotein (HDL) and Argonaute proteins (AGO, as demonstrated in number (4). Alternatively, they might be incorporated into multivesicular bodies (MVB) and secreted into exosomes; (2) pre-miRNAs are integrated into MVBs and released through exosomes, finishing their processing in target cells; (3) both pre-miRNA and mature miRNA can be integrated into microvesicles that sprout from the cell surface and reach the bloodstream.

**Figure 3 ijms-23-08653-f003:**
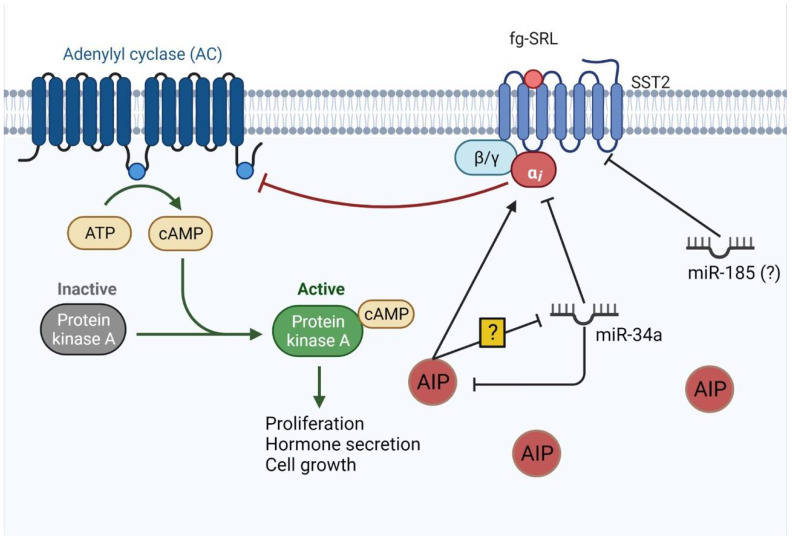
miRNAs involved in SST2 signaling. SST2 acts to inhibit adenylyl cyclase (AC) through inhibitory G protein. AIP also stimulates this inhibition. AC turns ATP into cyclic AMP (cAMP), which activates protein kinase A (PKA) that, through its pathway, leads to cell proliferation, growth, and hormone secretion. miR-34a targets AIP and G inhibitory a subunit and it is also inhibited by AIP by unknown mechanisms. miR-185 directly targets SST2 mRNA and might play a role in decreasing SST2 levels. Figure adapted from “Activation of Protein Kinase A (PKA)”, by BioRender.com (2022). Retrieved from https://app.biorender.com/biorender-templates.

**Table 1 ijms-23-08653-t001:** miRNAs’ expression associated with GH-secreting pituitary adenoma tumorigenesis.

Pathways/Proteins	miRNAs	miRNA Expression	Findings	References
PTTG	miR-126 miR-381 miR-338-3p	Downregulated Downregulated Upregulated	Proliferation Invasion	[54,55]
PTEN/PI3K/AKT/mTOR	miR-26b miR-128	Downregulated Upregulated	Tumor size Invasion	[55,56]
EMT	miR-503 miR-525-5p	Downregulated Upregulated	Differentiation, proliferation Proliferation, invasion, migration	[55]
HMGA1/2	miR-34b miR-548c-3p miR-34b miR-326 miR-432 miR-548c-3p miR-570	Downregulated	Proliferation	[57]
E2F1	miR-326 miR-603	Downregulated	Proliferation	[57]
RARS	miR-16-1	Downregulated	Inverse correlation with tumor size	[58]
FGF2	miR-503	Downregulated	Differentiation, proliferation	[55]

PTTG: pituitary-tumor-transforming gene; PTEN: phosphatase and tensin homolog; PI3K: phosphatidylinositol-4,5-biphosphate 3-kinase; AKT: PKB—protein kinase B; mTOR: mechanistic target of rapamycin; EMT: epithelial–mesenchymal transition; HMGA: high-motility group A; E2F1: E2 promoter binding factor 1; RARS: arginyl t-RNA synthetase; FGF2: fibroblast growth factor 2.

**Table 2 ijms-23-08653-t002:** miRNAs involved in response to fg-SRL.

miRNA	Target	miRNA Expression	Finding	Reference
miR-34a	AIP	Upregulated	Poor response to fg-SRL	[86,87]
miR-185	SST2	Upregulated	Poor response to fg-SRL	[89]
miR-125a-5p miR-524-5p	IGFBP-3 IGFALS chain precursor MMP-9	Downregulated	Better response to fg-SRL	[55]
miR-886-5p	P53/Bax pathway	Upregulated	Better response to fg-SRL	[55]

SRL: somatostatin receptor ligand; AIP: aryl hydrocarbon receptor-interacting protein; SST2: somatostatin receptor type 2; IGFBP-3: insulin-like growth factor binding protein 3; IGFALS: insulin-like growth factor binding protein acid labile subunit; MMP-9: matrix metalloproteinase-9; p53: tumor protein p53; Bax: Bcl-2 associated X, apoptosis regulator.

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
