# Peer review of "MicroRNA in Acromegaly: Involvement in the Pathogenesis and in the Response to First-Generation Somatostatin Receptor Ligands"

_ijms, 2022, doi:10.3390/ijms23158653_

Round 1

Reviewer 1 Report

The manuscript entitled ““MicroRNA in acromegaly: Involvement in the pathogenesis and in the response to first-generation somatostatin receptor ligands” by Henriques D. and colleagues, aims to provide a complete summary of the role of miRNA in the pathogenesis and in the drug-response of acromegaly. Overall, it is an interesting review that recapitulate the main passages in the miRNA biogenesis and their expression/role in tumorigenesis and treatment resistance in GH-secreting pituitary adenomas. Furthermore, the authors also provide a brief description of the principal circulating miRNA identified in acromegaly patients. The references used for this review are sufficiently updated and the paper is well organized and adequately described; however, some minor revision should be addressed:

-       First time introducing lanreotide (line 301) please specify that is a somatostatin analog drug.

-        In line 211 and 213 authors described results on cell lines AtT-20 and GH3, please specify/described what kind of cell lines are them (GH-secreting? ACTH-secreting? human or mouse)

-       The references used for this review are adequately updated (75%published in the last 10 years); however, I suggest to the authors to replace or confirm some of them (e.g., ref 22, 39, 44, 49 and 74) with more recent publications. The original articles were very important publications but not up to date enough in my opinion.

Reviewer 2 Report

I read with much interest the review submitted by Henriques et al. about miRNAs in acromegaly (pathogenesis, impact over sensitivity-to-SRL)

I do commend the authors for providing a clear review about a complex topic in the setting of a rare disease: references are properly refered, figures and tables help the readers to understand the subject and do not appear as being only figurative and, eventually, the manuscript is well written

I have only minor comments in the text :

l.39 : please add "namely octreotide and lanreotide" after somatostatin receptor ligands (fg-SRL)

l.42:  [..] its efficacy, in controlling the disease : please add "at the biochemical level" and remove the cut off for GH and IGF-1 proposed in brackets since they are not the ones consensually used to define biochemical remission

l. 43: [..] Pasireotide is a second-generation SRL multireceptor ligand. Please remove "multireceptor ligand" (pleonasm)

l. 50: [..] 4 to 12% of patients with apparently sporadic acromegaly, add "especially if they are young (age < 30 yo) and carry macroadenomas (Cuny et al. European Journal Endocrinology, 2013)"

l.69: [..] have been dedicated to these new molecules (plural)

l. 369 : "The acromegaly" : remove "The"

Reviewer 3 Report

In this study Henriques and Coll. report the available data about the role miRNAs expression in the pathogenesis of GH-secreting pituitary adenomas and in the resistance to first generation SSA.

The topic is very relevant and rather novel as applied to these tumors. The literature is fully analyzed and the miRNA world nicely described in the introduction.

My only concern is a somehow confused description of the role of some miRNA on pathways. For example, when reporting the modulatory role on PTEN/PI3K/Akt pathway, a clear distinction should be reported if miRNA target is PTEN (the inhibitor of the pathway, whose inhibition leads to Akt activation, or the other components (PI3K, Akt) whose downregulation leads to pathway activation. In this context the use of terms such as “regulation” should be avoided and replaced by “inhibition” or “activation”, for sake of clarity.

Moreover, Authors should clarify the apparent discrepancy between table 1, in which miRNA 26 is reported to target PTTG, and line 181 in which is reported as regulator od PI3K/Akt.
